# Prevalence and associated factors of depression among stroke family caregivers on follow up at Jimma medical center, southwest, Ethiopia: A cross-sectional study

Muhiddin Hirpasa Jabessa[1]*, Beshir Mammiyo Adem[2], Million Girma Tekle[2], Gutema Ahmed Fata[1], Hailemariam Hailesilasie Weldemariam[3]

1 Department of Psychiatry, College of Health Sciences, Jimma University, Jimma, Ethiopia, 2 Department of Psychiatry, College of Public Health and Medical ScieNces, Mettu University, Mettu, Ethiopia, 3 University of Bergen, Bergen, Norway

* muhiddinhirpasa@gmail.com

## Abstract

### Background

Taking care of stroke patients puts a high burden on their caregivers, and this leads to emotional disturbances like depression. However, little is known about the prevalence and associated factors of depression among family caregivers of stroke patients in Ethiopia. The aim of this study was to assess the prevalence and associated factors of depression among family caregivers of patients with stroke who have follow-up in Jimma Medical Center.

### Method

An institution-based cross-sectional study of 306 family caregivers of stroke patients was conducted using a consecutive sampling method. A structured interviewer-administered questionnaire was used to collect data. The Patient Health Questionnaires-9 (PHQ-9) was used to screen outcome variable (i.e., depression). The data was entered into Epi-data version 4.6 and exported to the statistical package for social science version 25 for analysis. Bivariable analyses followed by multivariable logistic regression models were performed. The association between depression and associated factors was estimated using an odds ratio of 95%CI and a p-value of less than 0.05 in the final model.

### Result

From a total 310 sample size 306 family caregivers were participated, making a response rate of 98.7%. The prevalence of depression among family caregivers was found to be 35.9% (95% CI = 30.54, 41.35). Poor social support (AOR = 2.31, 95%

**Data availability statement:** All relevant data are within the paper and its Supporting information files.

**Funding:** The authors received no specific funding for this work.

**Competing interests:** The authors have declared that no competing interests exist.

**Abbreviations:** ADHD, Attention Deficit Hyperactivity Disorder; ASD, Autism Spectrum Disorder; BDI, Beck Depression Inventory; BI, Barthel Index; CESD, Centers for Epidemiologic Studies Depression; CI, Confidence interval; DALYs, Disability Adjusted Life-Years; DSM-5, Diagnostic and Statistical Manual Fifth Edition; HADS, Hospital Anxiety and Depression Scale; HIV, Human Immunodeficiency Virus; JMC, Jimma Medical Center; MH, Muhiddin Hirpasa; OR, Odds Ratio; PHQ-9, Patient Health Questionnaire; QOL, Quality of Life; SPSS, Statistical Package for Social Science; SRQ-20, Standardized and Validated Self-Reported Questionnaire; TBI, Traumatic Brain Injury; WHO, World Health Organization; YLD, Years Lived with Disability.

CI: 1.05–5.08), severe physical dependence of patients (AOR = 3.09, 95% CI: 1.44–6.60), lack of medical insurance (AOR = 2.52, 95% CI: 1.34–4.75), spending more than 13 hours per day on care (AOR = 5.72, 95% CI: 2.32–8.12), and illness duration of 6 months (AOR = 3.50, 95% CI: 1.61–7.60) were factors significantly associated with depression.

## Conclusions and recommendations

This study found that more than one third of family caregivers of stroke patients have depression and almost the same with similar studies conducted across different countries of the world. Poor social support of caregivers, no health insurance by caregivers, greater than 13 hours of care per day, less than six months of illness duration, and severe dependence of patients on caregivers were variables associated with depression. Early detection of depression by health professional working at outpatient service and intervention of identified risk factors among stroke family caregivers were recommended for Jimma medical center and local health sector directors were recommended.

## Introduction

Globally, stroke has been detailed as one of the driving causes of mortality and disability [1]. Stroke has an impact both on the family caregivers and survivors [2]. It is a long-standing illness which has profound negative effect on their families [3]. After discharge from the hospital, about 45% of stroke patients will return to home, 24% will go to rehabilitation and 31% of them will be cared for by trained nurses [4].

Since most of the stroke patients spent most of their time at home, they need extensive physical and emotional care [5]. As a result, most caregivers suffer from emotional distress like depression, particularly during the acute stage of the illness [6].

Depression is defined as the most common form of mental illness in the general population; characterized as having cardinal symptoms either sadness, loss of interest or pleasure and four more symptoms from feelings of guilt or low self-worth, disturbed sleep or appetite, feelings of tiredness, and poor concentration and \or suicidal thought for at least two weeks [7].

Worldwide, the overall pooled prevalence of depressive symptoms among stroke family caregivers was 40.2% (95% confidence interval 30.1%–51.1%) [8].

Studies shows that over 70.4% of stroke patients are dependent on their caregivers and their level of dependency vary from moderate to severe [9]. Depression among caregivers not only affects the caregivers themselves, but also determines the care recipient's recovery, frequent hospital visits, and multiple admissions to the hospital [10]. So, the collaborative management of both the patients and family caregivers has a significant effect on the improvement and recovery of the patient [5].

Depression has the greatest negative effect on health status of the population either be alone or in associated with other chronic medical illnesses such as Asthma, Angina, and/or diabetes. The caregiver's related characteristics like being advanced

age, being female, being a spouse and presence of other chronic medical illness could increases the risk of depression among family caregivers [11].

Most of the time, caregivers have been considered as one of the overlooked population and this could result in poor health outcomes among caregivers [12]. It is reported that stroke patients' caregivers who developed depression and strain will have higher risk of death by 63% than general population over four years period [13].

Although depression is very common among stroke caregivers, to the best of our knowledge there is no literature done in our country up to this research was conducted. Thus, this study was aimed to assess the prevalence and associated factors of depression among family caregivers' of patients with stroke at Jimma Medical Center (JMC) to assess the burden of this problem in our country.

## Methods and materials

### Study area and design

Institution based a cross-sectional study was conducted at JMC from July 15– September 15 2021, south west, Ethiopia. Jimma is located in Oromia regional state and found 357 km away to South West of the capital city of Ethiopia, Addis Ababa. Jimma Medical Center (JMC) is a university teaching hospital and is the only referral hospital for the southwestern parts of the country. Currently, it provides service for more than 15 million people. The study was conducted at a chronic disease follow-up clinic among family caregivers of stroke patients. In chronic follow-up clinic of JMC, more than 3,000 patients consist of Hypertension, Diabetes Mellites, Epilepsy, Stroke, Thyroid dysfunction and other chronic medical illness, except HIV/AIDS getting service over a month. On average, 650 stroke patients visit chronic follow-up clinic over two months period.

### Source population and study population

All family caregivers (i.e., Parents, offspring, siblings, spouse, and other relatives) of stroke patients attending follow-up treatment at JMC during study period were source population. This includes any family caregivers who take a role in giving care for the stroke patients. Whereas, all consecutively selected family caregivers of stroke patient attending follow up treatment at JMC were study population. We included all family caregivers of stroke patients' attending follow-up at JMC and ≥ 18 years old and giving care for their relatives for at least two weeks. Whereas, family caregivers who were unable to respond due to illness and those with repeated visits (more than one visit) during data collection period were excluded to avoid repetition.

### Sample size and sampling technique

The sample size required for this study was calculated using single population proportion formula, by assuming "p" 24.3%, from the study conducted in Nigeria among stroke family caregivers [14]. Because we perceive that, the socio-economic status of this country is almost similar with ours. A 95% confidence interval (CI), 5% marginal of error (d) and 10% non-response rate were taken to calculate the sample size. The final sample size was 310. According to inclusion criteria participants were recruited for the data after patient's case folder was reviewed to check their diagnosis. The participants were selected consecutively for interview based on inclusion/exclusion criteria each day until the calculated sample size was reached. In case more than one family caregivers appear at a time, only one caregiver was selected by lottery method.

### Variables of the study

The dependent variable was depression among stroke family caregivers(yes/no). The independent variables incorporated in this study were include socio-demographic (Age, gender, religion, ethnicity, educational level, occupation, current residency, kinship relation to the patient and family income), psychosocial related variables (level of social support and time spent by family caregiver per day for caregiving) and Substance related variables (current use khat, alcohol and tobacco).

The independent variables specific to stroke patients include: socio-demographic factors (age, gender, religion, educational level, occupation and current residency), and clinical related variables (functional status \ severity of illness, duration of illness, presence of comorbid chronic physical illness and types of strokes).

## Data collection tools and procedure

Patients Hospital Case Folder: The patients' (stroke survivors) hospital case folder was used to obtain information about the clinical characteristics of the stroke survivors. Data was collected by structured and pre-tested interviewer administered questionnaires. It includes socio-demographic characteristics, clinical factors, current substance use, Patient Health Questionnaire-9 (PHQ-9) and oslo-3 social support scale. Data was collected by two Bachelor of Science in psychiatry and two Bachelor of Science in nursing after they have taken one day training on the data collection procedure. Supervised by two integrated clinical and community mental health master's students.

PHQ-9 was used to assess the level of depression among family caregivers and has nine items likert scale from zero to three. The tool score ranges from zero to 27 and used for dual purpose of screening and severity rating. Moreover, PHQ-9 has been validated in Ethiopian healthcare with sensitivity and specificity of 86.2% and 67.3% respectively at cut-off point ≥ 10 with internal consistency of Cronbach's alpha(0.81) [15]. The internal consistency of Cronbach's alpha of the tool from this study was 0.85.

Social support level of caregivers was assessed using Oslo 3-item Social Support (OSS-3) Scale. If the score is 3–8 it indicates poor social support, 9–11 indicates moderate social support, and 12–14 shows strong social support [16].

Modified Rankin scale (mRS) was used to assess the stroke severity/functional status of the stroke patients. It was developed in 1957 by Dr John Rankin in Glasgow, Scotland, the RS comprises 5 grades of stroke severity ranging from "no significant disability" to "severe disability). Strong inter-rater reliability($k = 0.76$) and Strong test-re-test reliability was also had been reported [17]. The specificity and sensitivity of the tool on caregivers of stroke patients at 3 and above cut-off point were reported to be 88.5% and 95.7% respectively [18].

## Data quality control

Questionnaires prepared in English were translated into local Languages Afan Oromo and Amharic by skilled professionals from language department. Then, back translated to English by independent skilled professionals who are blind for English version to check its consistency. Training was given for data collectors and supervisors by the principal investigator about the data collection. Pre-test was conducted among (5%) (n = 16) of the total sample size. Regular supervision was performed and filled questionnaires were checked for completeness and consistency.

## Data processing and analysis

The data were edited, cleaned, coded, and entered into the computer using Epi-Data version 4.6.2 and analyzed using Statistical Package for Social Sciences (SPSS) version 25. The presence of an association between dependent and independent variables was assessed using binary logistic regression analysis. Variables with a p-value less than 0.25 at bivariable binary logistic regression were entered into multivariable binary logistic regression. Backward variable selection method was used, in order not miss important variable. Statistical significance was considered at a p-value less than 0.05, and the strength of association was estimated by odds ratio at a 95% confidence interval. Descriptive statistics including frequencies, percentages, and mean were used to describe a finding. Chi-square and odds ratio were done to determine the association of variables. The model goodness of fit was checked by Hosmer and Lemeshow with value 0.47.

## Ethics approval and consent to participate

Ethical clearance was obtained from institutional ethical review board (IRB) of Jimma University Institute of Health Science with Ref. No: IHPGnD/362. The objective, purpose, and aims of the study were explained to each study participant.

Written informed consent was obtained from each of the participants prior to interview. Individuals who did not volunteer to participate on the interview were not forced to participate and well gone with thanks. Only assigned participants' identification number was used on questionnaire. Name of participants was neither mentioned during data collection nor entered into the computer for analysis. Participants who fulfilled for the depression according to assessment tool and had suicidal behavior were linked to psychiatry department for thorough evaluation and management.

## Results

### Socio-demographic characteristics of stroke patients and family caregivers

A total of 306 family caregivers were participated in this study making the response rate 98.7%. The mean age of family caregivers and patients in years were 34.03 ± 12.551 SD and 48.14 ± 14.751 SD respectively. Nearly two-third, 64.7% (n = 198) of family caregivers and more than half, 54.2% (n = 166) of stroke patients were males (Table 1).

### Clinical characteristics of stroke patients

The mean illness duration of the stroke patients participated in this study was 33.46 ± 30.987 SD in months. From the total stroke patients more than one third, 41.2%(n = 126) had comorbid Hypertension followed by Diabetes Mellitus 34.6% (n = 106) (See Table 2 below).

### Psychosocial related characteristics of family caregivers

The mean time spent per day in hour by caregiver with their patients was 6.34 ± 5.866 SD and nearly one third, 26.5% (n = 81) of the caregivers were spent more than 13 hours per day for caring their care recipients (Table 2).

### Substance related characteristics of family caregivers

From the total family caregivers about 46.4%(n = 142), 44.4%(n = 136) and 44.4% (n = 136) of them were used khat, tobacco and alcohol over the past three months respectively. Among users of khat, alcohol, and tobacco over the past three months, about 29.3% (n = 48), 29.4%(n = 50) and 44.9%(n = 61) were found to have depression respectively (Table 2).

### Prevalence and severity of depression in family caregivers

The result of current study showed that the prevalence of depression among family caregivers of stroke patients was found to be 35.9% (95% CI = 30.54, 41.35). Based on severity level, 16.4%(n = 51), 21.6%(n = 66), 13.1%(n = 40), 1.3%(n = 4) of the family caregivers have mild, moderate, moderately severe and severe form of depression respectively (Fig 1).

### Factors associated with depression

On multivariable logistic regression there were five variables which had significant association with stroke caregivers' depression. Those variables were; having poor social support of caregivers, severe physical dependence of patients, having no health insurance by caregivers, spending more than 13 hours per day for care and illness duration less than six months were found to have positive association with family caregivers' depression.

The result of this finding showed that family caregivers who had poor social support were 2.3 times more likely to develop depression than those with strong social support (AOR)= 2.31, 95% CI: 1.05–5.08). Caregivers with patient illness duration less than six months were 3.5 times more likely to develop depression than caregivers of patients' illness duration more than six months (AOR)= 3.50, 95% CI: 1.61–7.60). The odds of depression among family caregivers who spend more than thirteen hours per day for caring their relatives were nearly 5.7 times more likely as compared to those

**Table 1. Socio-demographic characteristics of family caregivers and patients with stroke follow up at Jimma medical center chronic follow up clinic, South-west Ethiopia 2021.**

| Variables | category | C (n) | C (%) | P (n) | P (%) |
|---|---|---|---|---|---|
| Age | 18-34 | 114 | 37.3 | 54 | 17.6 |
| | 35-54 | 80 | 26.1 | 146 | 47.8 |
| | ≥ 55 | 112 | 36.6 | 106 | 34.6 |
| sex | Male | 198 | 64.7 | 166 | 54.2 |
| | female | 108 | 35.3 | 140 | 45.8 |
| Marital status | Married | 122 | 39.9 | | |
| | single | 122 | 39.9 | | |
| | Separate/divorced/ widowed | 62 | 20.3 | | |
| Religion | Muslim | 188 | 61.4 | 178 | 58.2 |
| | orthodox | 76 | 24.8 | 86 | 28.1 |
| | protestant | 40 | 13.1 | 40 | 13.1 |
| | Others* | 2 | 0.7 | 2 | 0.7 |
| Current residency | Urban | 192 | 62.7 | 182 | 59.5 |
| | rural | 114 | 37.3 | 124 | 40.5 |
| Education | No formal ed | 99 | 32.4 | 198 | 64.7 |
| | primary (1–8) | 75 | 24.5 | 46 | 15 |
| | secondary (9–12) | 84 | 27.5 | 32 | 10.5 |
| | collage and above | 48 | 15.7 | 30 | 9.8 |
| occupation | Employed | 113 | 36.9 | 278 | 90.8 |
| | unemployed | 77 | 25.2 | 10 | 3.3 |
| | daily laborer | 42 | 13.7 | 4 | 1.3 |
| | Others** | 74 | 24.2 | 14 | 4.6 |
| Relationship | Parents | 46 | 15 | | |
| | siblings | 90 | 29.4 | | |
| | spouse | 71 | 23.2 | | |
| | offspring | 47 | 15.4 | | |
| | Other's relatives | 52 | 17 | | |
| Income | <2565 | 161 | 52.6 | | |
| | ≥2565 | 145 | 47.4 | | |
| Insurance | No | 138 | 45.1 | | |
| | yes | 168 | 54.9 | | |

Key= C-caregivers, p-patients, others*=catholic, wake feta, Jovan, others**=students, retired.

spend less hours per day (AOR)= 5.72, 95% CI: 2.32–14.12). Our study finding also revealed that family caregivers who have no health insurance were 2.5 times more likely to develop depression than those who have medical insurance (AOR)= 2.52, 95% CI: 1.34–4.75). It's also revealed that having more severe physical dependence care recipient increases odds of depression among family caregivers by three folds than having care-recipient with less severe physical dependence (AOR)= 3.09, 95% CI: 1.44–6.60) (Table 2).

## Discussion

According to our study the prevalence of depression among study participants was found to be 35.9% (95% CI:30.54, 41.35). This study finding was in line with other cross-sectional studies conducted in different countries. The prevalence

**Table 2. Bivariable and Multivariable binary logistic regression analysis for independent variables of depression among family caregivers of patients with stroke follow up treatment at Jimma medical center, southwest Ethiopia 2021.**

| Variables | Category | | Depression | | COR (95% CI) | P-values | AOR (95% CI) | P-values |
|---|---|---|---|---|---|---|---|---|
| | | | No N (%) | Yes N (%) | | | | |
| Age | 18-34 | | 80(70.2) | 34(29.8) | 1 | | | |
| | 35-54 | | 47(58.8) | 33(41.3) | 1.65(.91-3.0) | .101* | 1.25(.56-2.80) | .585 |
| | ≥55 | | 69(61.6) | 43(38.4) | 1.47(.84-2.56) | .175* | 1.65(.80-3.40) | .178 |
| Sex | Male | | 138(69.7) | 60(30.3) | 1 | | | |
| | Female | | 58(53.7) | 50(46.3) | 1.98(1.22-3.22) | .006* | 1.51(.76-3.00) | .243 |
| Marital status | Married | | 86(70.5) | 36(29.5) | 1 | | | |
| | Single | | 76(62.3) | 46(37.7) | 1.45(.85-2.47) | .176* | .622 (.27-1.45) | .272 |
| | Others | | 34(54.8) | 28(45.2) | 1.97(1.04-3.71) | .036* | .74(.33-1.68) | .472 |
| Monthly income | <2565 | | 100(62.1) | 61(37.9) | 1.20(.75-1.91) | .156* | 1.46(.78-2.75) | .240 |
| | ≥2565 | | 96(66.2) | 49(33.8) | 1 | | | |
| Duration in months | 0-6 | | 47(45.2) | 57(54.8) | 3.42(1.9-6.15) | **.000*** | **3.50(1.61-7.60)** | **.002**** |
| | 6-18 | | 70(71.4) | 28(28.6) | 1.17(.63-2.17) | .618 | 1.26(.55-2.88) | .586 |
| | ≥ 18 | | 79(76.0) | 25(24.0) | 1 | | | |
| Types of strokes | Ischemic | | 84(47.7) | 92(52.3) | 1.97(1.22-3.20) | .006* | 1.70(.92-3.14) | .092 |
| | hemorrhagic | | 112(86.2) | 18(13.8) | 1 | | | |
| Level of dependence | Mild | | 84(75.0) | 28(25.0) | 1 | | | |
| | moderate | | 70(74.5) | 24(25.5) | 1.08(.57-2.03) | .816 | 1.00(.45-2.23) | .994 |
| | severe | | 42(42.0) | 58(58.0) | 2.93(1.65-5.22) | **.000*** | **3.09(1.44-6.60)** | **.004**** |
| Time spent per day in hrs. | 1-4 hrs | | 61(78.2) | 17(21.8) | 1 | | | |
| | 5-8 hrs | | 58(76.3) | 18(23.7) | 1.11(.52-2.37) | .780 | .583(.24-1.58) | .289 |
| | 9-12 hrs | | 48(67.6) | 23(32.4) | 1.72(.83-3.58) | .147* | 1.14(.43-2.97) | .797 |
| | 13 and above hrs | | 29(35.8) | 52(64.2) | 6.43(3.18-13.00) | **.000*** | **5.72(2.32-14.12)** | **.000**** |
| Medical insurance | No | | 70(50.7) | 68(49.3) | 2.3(1.40-3.72) | **.001*** | **2.52(1.34-4.75)** | **.004**** |
| | yes | | 126(75) | 42(25) | 1 | | | |
| Social support | Poor | | 59(47.6) | 65(52.4) | 2.57(1.45-4.54) | **.001*** | **2.31(1.05-5.08)** | **.038**** |
| | Moderate | | 72(80) | 18(20) | .59(.30-1.18) | .134* | .640(.26-1.60) | .341 |
| | Strong | | 65(70.7) | 27(29.3) | 1 | | | |
| Current khat use | No | | 116(70.7) | 48(29.3) | 1 | | | |
| | Yes | | 80(56.3) | 62(43.7) | 1.86(1.16-2.97) | .010* | 1.80(.96-3.36) | .065 |
| Current alcohol use | No | | 120(70.6) | 50(29.4) | 1 | | | |
| | Yes | | 76(55.9) | 60(44.1) | 1.57(.98-2.51) | .061* | 1.67(.90-3.12) | .107 |
| Current tobacco use | No | | 121(71.2) | 49(28.8) | 1 | | | |
| | Yes | | 75(55.1) | 61(44.9) | 1.43(.89-2.28) | .137* | 1.36(.73-2.53) | .334 |
| Comorbid physical illness | Kidney problem | No | 160(68.4) | 74(31.6) | | | | |
| | | Yes | 36(50.0) | 36(50.0) | 1.75(1.02-2.97) | .041* | 1.53(.75-3.13) | .242 |
| | DM | No | 138(69.0) | 62(31.0) | 1 | | | |
| | | Yes | 58(54.7) | 48(45.3) | 1.84(1.13-2.99) | .014* | 1.64(.84-3.20) | .146 |
| | HTN | No | 120(66.7) | 60(33.3) | 1 | | | |
| | | Yes | 76(60.3) | 50(39.7) | 1.17(.73-1.88) | .113* | 1.56(.82-2.96) | .177 |
| | Epilepsy | No | 146(66.4) | 74(33.6) | 1 | | | |
| | | Yes | 50(58.1) | 36(41.9) | 1.05(.63-1.74) | .166* | .58(.29-1.13) | .110 |

Key = others(marital)= widowed, separated, divorced, * = p-value < 0.25 statistically significant at bivariate, ** = p-value < 0.05 statistically significant at multivariate, 1 = reference category. Hosmer and Lemeshow test: 0.47.

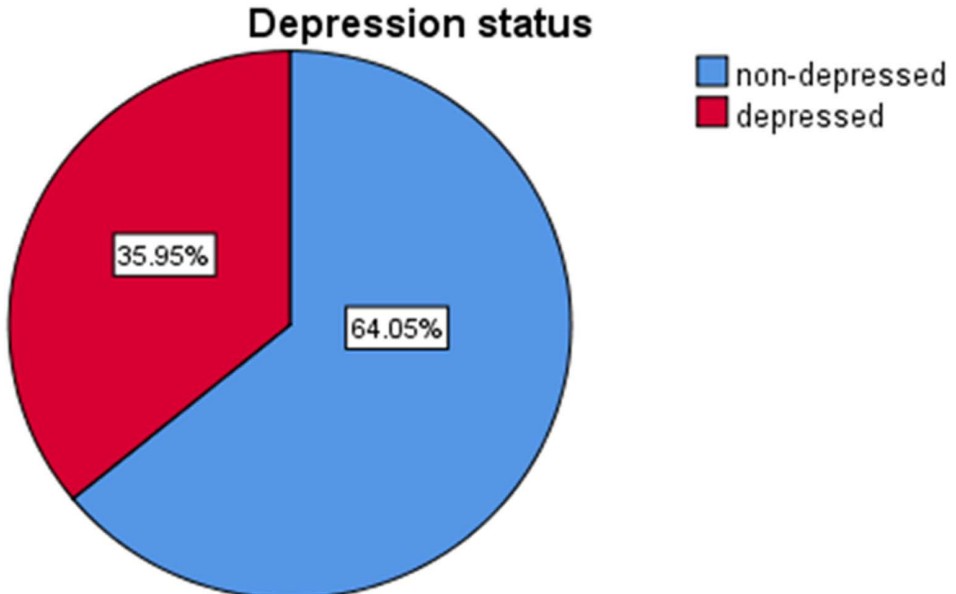

**Fig 1. (A) The red color of pie chart indicates caregivers with depression, while blue color represents those who are non-depressed.** (B) The number on each color of pie chart indicate the percentage of caregivers' depression 35.9% (n = 110) and non-depressed 64.05%(n = 196) at 95% CI based on PHQ-9 screening tool.

of depression was 37% in India [19], 39% in UK [20], in Finland ranging (30%– 33%) [5] and among Iranian was reported as 40% [21]. The Global pooled prevalence estimates the prevalence of depression among stroke family caregivers to be 40.2% [8].

However, the prevalence of depression in this study was lower than cross-sectional study done in Malaysia (63.8%). The study was conducted during six-month follow up period, and participants were assessed using PHQ-9 [22]. The discrepancy might be because of difference in socio-economic status. The other supporting reason might be in that, the study was conducted among family caregivers of stroke patient whose illness duration was less than six months in which the depression status increased during the acute phase of the illness [23]. Another cross-sectional study conducted in China found prevalence of depression 44.6% among 92 stroke family caregivers based on BDI assessment scale [2]. This might be due to different assessment tools used; in that BDI scale was used, in which overestimation of depression was reported with this tool. High level female gender respondent, in which females were more vulnerable to depression than male [4]. Similarly, a cross-sectional study done in Japanese reported magnitude of depression using GDS scale was found to be higher (52%) than this study result [24]. The discrepancy of the results might be in that majority, 191(78.6%) of the participants were female [4] and spouse 116(47.7%) in which higher depression was reported among these population than their counter parties [23].

On the other hand, our study result was found to be higher than similar studies design conducted in Nigeria and India, where both used HADS assessment scale and found prevalence of depression as 18% and 24.3% and 18%(18) [14] respectively. The difference in the results could be attributed to the smaller sample size they used and the difference in assessment tools. Obviously it's known that whenever the sample size become it might over/underestimate the finding. Another cross-sectional study conducted in Brazil by using HADS scale, the prevalence of depression was lower (22.6%) than our study finding [25]. The first reason for the difference might be difference in health service being delivered and difference in socio-economic status. The other possible reasons may be lower level of sample size used.

Similarly, having severe physically dependent care recipients increases the odds of depression more than three times among their family caregivers than their counter parties. This finding was in line with study done in Iran [21], UK university of Glasgow [26,27], Finland [5] and America [28]. This might be explained as care recipients who are severely dependent on their caregivers may have no time for themselves and all responsibilities would be on their shoulder and this impose to family caregivers emotional disturbances [2].

In addition, family caregivers who spend more hours per day for giving a care had nearly six times odds of depression than those spend less hours per day. This is supported by studies done in Japanese [24,29] and China [30]. The possible reason for this might be family caregivers who spend more time may occupy too much private time of themselves and thus reduces their work, social participation, sleep and entertainment time [30].

Duration of the illness is also one of the associated factors that increase caregiver's depression vulnerability by more than three folds. Similar finding were reported from studies done in India [31],China [32] and Pakistan [23]. This is probably because of unpreparedness following transition from hospital to home, dealing with new roles and these may leads to negative mood state [33]. The other possible reasons may include the decreasing stroke severity after period of time [28] and increased recovery of the illness with time change [34].

Health insurance was also one of the factors that associated with caregiver depression and those family caregivers with no health insurance has 2.5 times more likely to develop depression than their counter parties. This is supported by the findings from Uganda [35], China [30], Korea [36] and UK [37]. This might best explained as the economic burden is one of the main factor for the emotional disturbances among caregivers [35].

Furthermore, having poor social support by caregivers increases the odds of depression by 2.3 folds among family caregivers. This is supported by different findings from the literature; Ghana [38], Korea [36], Malaysia [39] and UK [37]. This is likely due to the fact that having social support reduces the perceived amount of stress in a difficult scenario and boosts caregivers' coping capacity during emotional disturbance [40].

## Limitation of the study

Since the study was interviewer-administered, social desirability bias was a potential concern. Additionally, the facility-based nature of the study may limit its generalizability. Being a cross-sectional study method makes difficult to know cause effect relationship between outcome and predictors.

## Conclusion

This study found that the prevalence of depression among family caregivers of stroke patients was almost in similar range with the pooled prevalence of world report. According to the current study's findings; having poor social support, having no medical insurance, spending greater than 13 hours to provide care per day, having duration illness less than six months and being caregiver's of severely dependent patients were variables that increases risk of developing depression among participants.

## Supporting information

**S1 File. Mud spss coded.**
(SAV)

## Author contributions

**Conceptualization:** Muhiddin Hirpasa Jabessa, Hailemariam Hailesilasie Weldemariam, Million Girma Tekle, Beshir Mammiyo Adem.

**Data curation:** Gutema Ahmed Fata, Million Girma Tekle.

**Formal analysis:** Hailemariam Hailesilasie Weldemariam.

**Funding acquisition:** Gutema Ahmed Fata.

**Investigation:** Muhiddin Hirpasa Jabessa.

**Methodology:** Muhiddin Hirpasa Jabessa, Hailemariam Hailesilasie Weldemariam.

**Project administration:** Million Girma Tekle.

**Resources:** Muhiddin Hirpasa Jabessa.

**Software:** Muhiddin Hirpasa Jabessa.

**Supervision:** Beshir Mammiyo Adem.

**Validation:** Muhiddin Hirpasa Jabessa, Gutema Ahmed Fata.

**Visualization:** Hailemariam Hailesilasie Weldemariam, Gutema Ahmed Fata, Million Girma Tekle, Beshir Mammiyo Adem.

**Writing – original draft:** Muhiddin Hirpasa Jabessa.

**Writing – review & editing:** Muhiddin Hirpasa Jabessa.

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
