## [Decision Letter · Decision Letter 0]

13 Sep 2022

Dear Dr. Jabessa,

Thank you for submitting your manuscript to PLOS ONE. After careful consideration, we feel that it has merit but does not fully meet PLOS ONE’s publication criteria as it currently stands. Therefore, we invite you to submit a revised version of the manuscript that addresses the points raised during the review process.

We look forward to receiving your revised manuscript.

Kind regards,

Jamie Males

Editorial Office

PLOS ONE

A clean copy of the edited manuscript (uploaded as the new *manuscript* file).

“NO”

4. Thank you for stating the following in the Funding Section of your manuscript:

“Jimma medical center has sponsored for cost of data collection.”

“No”

Reviewers' comments:

Reviewer's Responses to Questions

**Comments to the Author**

1. Is the manuscript technically sound, and do the data support the conclusions?

Reviewer #1: Partly

2. Has the statistical analysis been performed appropriately and rigorously?

Reviewer #1: Yes

3. Have the authors made all data underlying the findings in their manuscript fully available?

Reviewer #1: Yes

4. Is the manuscript presented in an intelligible fashion and written in standard English?

Reviewer #1: No

Reviewer #1: The comments

It is a great honor to review the manuscript in this well-recognized journal. I am delighted.

Then I have some comments as follows

The paper is nice and tried to assess an important topic. But I have some comments.

1. The author did not make it clear who are the caregivers. The family? The wives? The paid nurses? Or others, and how did the author get those care givers….all accompanying individuals to the hospitals are not the caregivers….

2. PHQ 9 is meant for screening purposes, not for diagnosis. Then how the diagnosis of depression was made?

3. low social support is considered a predictor of depression, but is it for stroke patients or for caregivers?

4. having no health insurance is again a predictor of depression? This is not clear. Paid caregivers or family members?

6. The author did not narrate the details of the caregivers, are they professionals?... nurses,,,,

**Do you want your identity to be public for this peer review?** For information about this choice, including consent withdrawal, please see our Privacy Policy

Reviewer #1: No

---

## [Author Response · Author response to Decision Letter 1]

1 Oct 2022

Rebuttal letter

Dear Editor,

We are grateful to the reviewer for the time taken to review the manuscript, and the useful comments they provided to improve the manuscript. We have responded to the entire reviewer’ comments point-by-point, highlighting the changes in the manuscript; and they are summarized below: Once again, we thank the reviewers for the useful comments and suggestions.

Comments of the reviewers Reply by the author(s)

Editor #1

1.Please ensure that your manuscript meets PLOS ONE’s style requirements, including those for the file naming.

The comment accepted and corrected accordingly.

2. we suggest you thoroughly copyedit your manuscript for language usage, spelling and grammar.

Thank you for the comment raised. We try to correct spelling error and grammar for language usage as highlighted in the manuscript by consulting different language experts.

3. relate with funding

a. please clarify the source of funding (financial or material support) for your study.

b. state what role the funder took in the study. If the funder had no role in your study please state “the funders had no role in study design, data collection and analysis, decision to publish, or preparation of the manuscript”

c. if any authors received a salary from any of your funders, please state which authors and which funders.

d. If you did not receive any funding for this study, please state: “the authors received no specific funding for this work.” Comment accepted and corrected accordingly.

4.funding should not appear in the acknowledgement section or other areas of your manuscript.

Comment accepted and corrected accordingly. We have removed funding related statements from the whole manuscript and included in the cover letter.

5. we noted that you have stated that you will provide repository information for your data at acceptance. Should your manuscript be accepted for publication, we will hold it until you provide the relevant accession numbers or DOIs necessary to access your data.

Thank you for providing this information and we have already summited repository data, so that you can accept it for publication as it was mentioned in cover letter.

6. your ethics statement should only appear in the method section of your manuscript

Thank you for the comment and it is corrected accordingly.

Reviewer # 1

1.The author did not make it clear who are the caregivers, the family? The wives? The paid nurses Or others, and how did the author those caregivers…. all accompanying individuals to the hospitals are not the caregivers…

Thank you for the comment raised by the reviewer.

The comment accepted and corrected accordingly.

2.PHQ 9 is meant for screening purposes, not for diagnosis. Then how the diagnosis of depression was made?

Actually, PHQ 9 used for screening purpose and severity rater and the comment accepted and corrected accordingly. But when we’re reporting the caregiver’s depression, we’re not saying depression disorder, but based on assessment tool those scored 10 and above were considered as depressed which doesn’t mean that they have depression disorder. This is also supported by the literatures

3. poor social support is considered a predictor of depression, but is it for stroke patients or caregivers?

There might be an association between patients’ poor social support and caregiver depression, but in our study it caregivers poor social support was a predictor of caregivers’ depression

4. Having no health insurance is again a predictor of depression. This is not clear. Paid caregivers or family members. As our study population was stroke family caregivers, we’re stating that having no health insurance was a predictor of caregivers’ depression.

5. The author did not narrate the details of caregivers are they professionals?... nurse...

The comment accepted and corrected accordingly.

---

## [Decision Letter · Decision Letter 1]

12 Dec 2022

Dear Dr. Jabessa,

Thank you for submitting your manuscript to PLOS ONE. After careful consideration, we feel that it has merit but does not fully meet PLOS ONE’s publication criteria as it currently stands. Therefore, we invite you to submit a revised version of the manuscript that addresses the points raised during the review process.

We look forward to receiving your revised manuscript.

Kind regards,

Saurav Basu, M.D.

Academic Editor

PLOS ONE

Reviewers' comments:

Reviewer's Responses to Questions

**Comments to the Author**

Reviewer #1: All comments have been addressed

Reviewer #2: All comments have been addressed

2. Is the manuscript technically sound, and do the data support the conclusions?

Reviewer #1: Yes

Reviewer #2: Yes

3. Has the statistical analysis been performed appropriately and rigorously?

Reviewer #1: Yes

Reviewer #2: Yes

4. Have the authors made all data underlying the findings in their manuscript fully available?

Reviewer #1: Yes

Reviewer #2: Yes

5. Is the manuscript presented in an intelligible fashion and written in standard English?

Reviewer #1: Yes

Reviewer #2: Yes

Reviewer #1: thank you for response. the authors almost responded for the concerns but I have some questions.

what is the practice of health insurance in your set up?

is there any care give who get paid?

Reviewer #2: The study has a focus; however, it lacks clarity on multiple dimensions (differentiation between variables relevant to stroke caregivers vs those of stroke survivors, page 1, keywords), inclusion/exclusion criteria, lines 95-98, 115,116, sampling techniques, lines 94-104). The data collection tool is a combination of multiple scales that adress both stroke survivors and stroke caregivers as well which creates another site of confusion, lines 111-113. Finally, an in-depth proof reading is required to ensure English language errors are fixed, lines 69, 70,71,72,86.

**Do you want your identity to be public for this peer review?** For information about this choice, including consent withdrawal, please see our Privacy Policy

Reviewer #1: No

Reviewer #2: **Yes: ** Nariman Ghader

---

## [Author Response · Author response to Decision Letter 2]

26 Dec 2022

Rebuttal letter

Dear Editor,

We are grateful to the reviewers for the time taken to review the manuscript, and the useful comments they provided to improve the manuscript. We have responded to the entire reviewer’ comments point-by-point, highlighting the changes in the manuscript; and they are summarized below: Once again, we thank the reviewers for the useful comments and suggestions.

Comments of the reviewers and editors Reply by the author(s)

Editor #1

1 Your sampling methodology is unclear on whether multiple participants (family members) were selected from the same patient household/family. You need to clarify accurately and clearly.

I also recommend a language and grammar check.

Reply by the author(s)

Thank you for your constructive comments.

The comment is accepted and corrected accordingly.

Regarding the case in which multiple caregivers were appear together, only one caregiver was selected based on lottery mothed, as it can be found on the manuscript. Page # 5, line 111,112.

Reviewer #1

Reply by the author(s)

Thank you for response. the authors almost responded for the concerns but I have some questions:

1. What is the practice of health insurance in your set up?

#response by author(s)

Here, in our setup the purpose of health insurance is that; those with health insurance will have free services like: medication, hospital admission bed, and other payment related services.

As economic burden is one of the risk factors for stroke caregiver depression, those with health insurance were less vulnerable for depression. So that, health insurance was considered as one of independent variable in our study as it can be supported by different literature.

2. Is there any care giver who get paid?

response by author(s)

As it is mentioned on the response for question # 1, our study was on stroke family caregivers and there is no any payment for the caregiving service of their relatives. And our study participants were limited to stroke family caregivers.

Reviewer #2: Thank you for your constructive comments. The comment was accepted and corrected accordingly.

1.The study has a focus; however, it lacks clarity on multiple dimensions (differentiation between variables relevant to stroke caregiver’s vs those of stroke survivors, page 1, keywords),

inclusion/exclusion criteria, lines; 95-98, 115,116, sampling techniques, lines 94-104).

response by author(s)

#Regarding the sample size calculation, we took “p” 24.3%, from the study conducted among stroke family caregivers in Nigeria. Because we perceived that, the socio-economic status of this country is almost similar with ours.

#The variables were separated as that of patients and caregivers.

# Regarding sampling techniques, the comment was accepted and corrected accordingly.

2. The data collection tool is a combination of multiple scales that address both stroke survivors and stroke caregivers as well which creates another site of confusion, lines 111-113. The comment was accepted and corrected accordingly.

The data collection has discussed for the patients and caregivers separately.

3. Finally, an in-depth proof reading is required to ensure English language errors are fixed, lines 69, 70,71,72,86. The comment was accepted and corrected accordingly.

---

## [Decision Letter · Decision Letter 2]

5 Apr 2023

Dear Dr. Jabessa

Thank you for submitting your manuscript to PLOS ONE. After careful consideration, we feel that it has merit but does not fully meet PLOS ONE’s publication criteria as it currently stands. Therefore, we invite you to submit a revised version of the manuscript that addresses the points raised during the review process.

We look forward to receiving your revised manuscript.

Kind regards,

Saurav Basu, M.D.

Academic Editor

PLOS ONE

Journal Requirements:

Reviewers' comments:

Reviewer's Responses to Questions

**Comments to the Author**

Reviewer #3: All comments have been addressed

2. Is the manuscript technically sound, and do the data support the conclusions?

Reviewer #3: Partly

3. Has the statistical analysis been performed appropriately and rigorously?

Reviewer #3: Yes

4. Have the authors made all data underlying the findings in their manuscript fully available?

Reviewer #3: Yes

5. Is the manuscript presented in an intelligible fashion and written in standard English?

Reviewer #3: Yes

Reviewer #3: Thank you very much for providing this updated version of the article.

In paragraph Data processing and analysis, please explain with more detail how you built the muktivariate regression model, was it backward or forward?

In the Discussion section, please provide the limitations due to the sampling (facility-based), how Berkson's bias can be present in the study and the overall effect on the OR estimation that this study provides, compared to a population-based study.

**Do you want your identity to be public for this peer review?** For information about this choice, including consent withdrawal, please see our Privacy Policy

Reviewer #3: No

---

## [Author Response · Author response to Decision Letter 3]

9 May 2023

Reviewer # 3

1.Inparagraph data processing and analysis, please explain with more detail how you built the multivariate regression model, was it backward or forward?

Thank you for your constructive comments.

The comment was accepted and corrected accordingly.

2. In the Discussion section, please provide the limitations due to the sampling (facility-based), how Berkson's bias can be present in the study and the overall effect on the OR estimation that this study provides, compared to a population-based study.

That’s good and constructive comments that, as our study design is cross-sectional we can’t generalize about cause-effect relationship as compered to berkson’s bias on case control group(clinical trial).

In addition to what has been mentioned in the body of manuscript, the facility that we had took the sample is giving service for more than fifteen millions of individuals from different classification and its more representative.

---

## [Decision Letter · Decision Letter 3]

12 Sep 2023

Dear Dr. Jabessa,

Thank you for submitting your manuscript to PLOS ONE. After careful consideration, we feel that it has merit but does not fully meet PLOS ONE’s publication criteria as it currently stands. Therefore, we invite you to submit a revised version of the manuscript that addresses the points raised during the review process.

We look forward to receiving your revised manuscript.

Kind regards,

Nega Degefa Megersa, msc

Academic Editor

PLOS ONE

Journal Requirements:

Additional Editor Comments:

1. Avoid italicizing your text in the abstract section unless particularly requested.

2. Consider the proper and consistent use of analysis method in your study. I think the difference between multivariate and multivariable logistic regression analysis be noted and used accordingly.

3. Line (30): you can make it, the association between depression and associated factors was estimated using odds ratio with its 95%CI and p-value.

4. Line (32): a total of 306 family caregivers participated in the study, making a response rate of 98.7%.

5. Line (39): it is good to put the meaning of your finding in the conclusion. What does more than 1/3rd prevalent mean in the study’s context. It is also good to highlight the implication of this finding to the study’s context.

6. Line 57: It is a long-standing and weakening condition that has a profound negative effect on their families. It looks grammatically incorrect, please make sure the sentence is coherent with preceding one.

7. Line72: Although depression is very common among stroke caregivers, little is found in literature. Since most stroke patients spend most of their time at home, they need extensive physical and emotional care. Correct as underlined and red marked in the sentence.

Line 76: The finding revealed that every year around 33% of people left uunemployed due to depression, more than non-depressed. Correct the spelling as unemployed. The background doesn’t adequately address context of the problem in an orderly pattern (from global, to local)

Line 83: So, our study was aimed to assess the prevalence and associated factors of depression among study participants at specified area. Put the area instead.

Line 81: Worldwide, the overall pooled prevalence of depressive symptoms among stroke family caregivers was 40.2%, this is a report from a systematic review, which seems inconsistent with the idea on line 72, which states that little is found on literature (Although depression is very common among stroke caregivers, little is found on literature). So, I advise the authors to resolve this inconsistency.

Line (168): Chi-square and odds ratio were done to determine the association of variables. The model goodness of fit was checked by Hosmer and Lemeshow.

Suggestion to improve clarity.

The association between the outcome and explanatory variables were measured using Chi-square and odds ratio.

Please specify the value of the model fitness test, to show whether the model adequately fits the data.

Line (184): Avoid reciting referenced table by using indicative terms like see table1. I think it is enough to cite the table or figure like (table 1) or (figure 1).

Line (77): I think you need to extend you’re the limitation of your study by adding the nature of your sampling method (consecutive sampling) and the generalizability of your study as well as the geographical limitation of your study which is not representative of the wider context.

Line (82): suggestion to improve your conclusion.

Make sure your conclusion captures the interpretation and the possible implication of the finding instead of putting the value as indicated in the result. Instead of mentioning as more than one third which is vague, you can present the finding as high, consistent or low relative to the previous study, which highlights the trend and relative importance of the finding.

Reviewers' comments:

Reviewer's Responses to Questions

**Comments to the Author**

Reviewer #4: (No Response)

2. Is the manuscript technically sound, and do the data support the conclusions?

Reviewer #4: Partly

3. Has the statistical analysis been performed appropriately and rigorously?

Reviewer #4: Yes

4. Have the authors made all data underlying the findings in their manuscript fully available?

Reviewer #4: Yes

5. Is the manuscript presented in an intelligible fashion and written in standard English?

Reviewer #4: No

Reviewer #4: Title: Prevalence and associated factors of depression among stroke family caregivers on follow up at Jimma medical center, southwest, Ethiopia: A cross-sectional study

Thank you for considering me to review this manuscript. The paper raises a significant problem since caretakers pass through several challenges, including mental issues, while caring for patients with stroke. I put my concerns as follows:

Abstract

The background section of the abstract must show the burden of the problem, the gap, and the intention of this study. You can merge the objective here.

Add the study period under the method section. Appropriately use words like "multivariate and multivariable." The sample size you put under the method and result sections are the same; if so, why is the response rate 98.7%?

Correct the upper limit of the confidence interval for the factor "spending more than 13 hours per day on care (AOR = 5.72, 95% CI: 2.32-1.12)."

Introduction

The introduction section lacks critical components, and intensive revisions are mandatory. The figures (prevalence of depression among caretakers of stroke patients) from previous studies must be addressed in a global-to-local context approach. Keep coherence and avoid contradicting ideas. For example, in line 72, you write, "…little is found on literature…" On the other hand, you put the pooled prevalence of depression from lines 81 to 83. In addition, the risk factors are not well addressed. Furthermore, some information is about stroke patients and others about their caretakers; be focused on your title.

The risk factors of depression among caretakers must also be included,and add citations appropriately. Previously proposed solutions at the national level or in any means, if any, should be included. The limitations of previously published studies and the gaps identified must also be included.

Furthermore, it is better to start with the global and national burden and impacts of stroke in different dimensions, including the economic consequences.

Methods

There are critical problems in this section. The source and study population are not appropriately defined. You write parents, offspring, siblings, spouse, and other relatives are the sources population, and if you get two during the visit, you select one of them by lottery method. Imagine if the stroke patient is married and his partner and his relative come together for the follow-up visit their relative has a 50% chance to be included in this study. I think this is unacceptable because his partner lives with him and she suffers a lot more than the relative. So, from the outset, the lottery method must be for equal relationships; otherwise, the findings are either over or underestimated. You tried to see it as a factor; however, the sampling approach is unacceptable.

Sampled and study populations are different; hence, you must modify your study population.

Appropriately organize the headings and the information within each header. For example, the information under the data collection and procedure from lines 131 to 149 does not align with the title. You have to modify it with other appropriate headings. Moreover, how did you organize your tool? Several missed variables may affect depression, so why do you extensively include them from the outset? Some variables were not appropriately measured, e.g. How did you assess alcohol use? In general, the method section needs further detailed clarification.

Results and discussion

Paraphrase the sentences and appropriately link the tables and figures; do not write as see Table 1 below; instead, put like (Table 1). Where is figure one?

Modify fragmented paragraphs under the discussion section. The majority of the references are outdated; please replace them with recent ones.

**Do you want your identity to be public for this peer review?** For information about this choice, including consent withdrawal, please see our Privacy Policy

Reviewer #4: No

---

## [Author Response · Author response to Decision Letter 4]

14 Sep 2023

1. Avoid italicizing your text in the abstract section unless particularly requested.

Thank you for your constructive comments.

The comment was accepted and corrected accordingly.

2. Consider the proper and consistent use of analysis method in your study. I think the difference between multivariate and multivariable logistic regression analysis be noted and used accordingly.

That’s good and constructive comments.

We have corrected accordingly and it’s highlighted

3. Line (30): you can make it, the association between depression and associated factors was estimated using odds ratio with its 95%CI and p-value. It's modified accordingly.

4.Line (32): a total of 306 family caregivers participated in the study, making a response rate of 98.7%.

The total sample size of this study was 310 as it’s mentioned on method part of the documents. Of which only 306 was participated in the study which make response rate 98.7%.

5. Line (39): it is good to put the meaning of your finding in the conclusion. What does more than 1/3rd prevalent mean in the study’s context. It is also good to highlight the implication of this finding to the study’s context.

That’s good comment and we have modified accordingly

6. Line 57: It is a long-standing and weakening condition that has a profound negative effect on their families. It looks grammatically incorrect, please make sure the sentence is coherent with preceding one.

Modified and highlighted

8.the other comments Other comments are modified accordingly

9.The background section of the abstract must show the burden of the problem, the gap, and the intention of this study. Thank you for your constructive comments dear reviewer.

# As the abstract section of the manuscript shows the summary, we only included the world report as the burden of the problem. And we merge the background and objective of the study as highlighted

10. The sample size you put under the method and result sections are the same; if so, why is the response rate 98.7%?

Correct the upper limit of the confidence interval for the factor "spending more than 13 hours per day on care (AOR = 5.72, 95% CI: 2.32-1.12)."

Introduction

The total sample size was 310 as mentioned in the method part. Out of which 306 were involved in the study which make the response rate 98.7%

Comment accepted and corrected accordingly

11.The introduction section lacks critical components, and intensive revisions are mandatory. The figures (prevalence of depression among caretakers of stroke patients) from previous studies must be addressed in a global-to-local context approach.

we accepted the comments and addressed it accordingly. Regarding the magnitude of the problem, we have mentioned the global prevalence as the summary of the report. Otherwise to the best our knowledge there is no local finding and, even in the African continent only 1 report we had found and mentioned in discussion part.

We also try to address risk factors we got from literature and scientific finding, which doesn’t mean that all risk factors are addressed. As its difficult to address all at once additional study might be indicated making this as source. Otherwise, other comments regarding grammatical errors and ideas out of our title has been omitted

#11.The source and study population are not appropriately defined. You write parents, offspring, siblings, spouse, and other relatives are the sources population, and if you get two during the visit, you select one of them by lottery method. Imagine if the stroke patient is married and his partner and his relative come together for the follow-up visit their relative has a 50% chance to be included in this study. I think this is unacceptable because his partner lives with him and she suffers a lot more than the relative. So, from the outset, the lottery method must be for equal relationships; otherwise, the findings are either over or underestimated. You tried to see it as a factor; however, the sampling approach is unacceptable.

Sampled and study populations are different; hence, you must modify your study population. #Source population is to whom we generalize our finding and the study population is from which we select our study participant, thus we define accordingly.

# The reason why we used lottery method was just to avoid over/underestimation of our finding. If we select the one who spent more time with the patient obviously it will overestimate the finding, so this was the only chance we had to use.

# as mentioned earlier JMC was one of the organization giving service for more than fifteen million diversified population living in south western part of the country. This will somewhat representative, even though not as that of national level-based study.

#12. For example, the information under the data collection and procedure from lines 131 to 149 does not align with the title. You have to modify it with other appropriate headings. Moreover, how did you organize your tool? Several missed variables may affect depression, so why do you extensively include them from the outset? Some variables were not appropriately measured, e.g. How did you assess alcohol use? In general, the method section needs further detailed clarification.

Comments accepted and addressed accordingly

# comments regarding tools we have mentioned the psychometric properties of the tools we have used including the validation. But what we have mentioned in this manuscript was those variables statistically significant at binary multivariable regression model. Example alcohol, other substances were measured using adopted ASSIST substance use scale. Since substance were not significant according to our study we didn’t mention it in the manuscript. Otherwise, other comments are modified accordingly.

#13. Paraphrase the sentences and appropriately link the tables and figures; do not write as see Table 1 below; instead, put like (Table 1). Where is figure one?

Modify fragmented paragraphs under the discussion section. The majority of the references are outdated; please replace them with recent ones

Comment accepted and modified

Regarding the reference we are obligated to use out of dated reference since we didn’t found literature inline with our study objective/title

---

## [Editor Report · Decision Letter 4]

19 Sep 2023

Dear Dr. Jabessa, 

We look forward to receiving your revised manuscript.

Kind regards,

Nega Degefa Megersa, msc

Academic Editor

PLOS ONE

Journal Requirements:

Please review your reference list to ensure that it is complete and correct. If you have cited papers that have been retracted, please include the rationale for doing so in the manuscript text or remove these references and replace them with relevant current references. Any changes to the reference list should be mentioned in the rebuttal letter that accompanies your revised manuscript. If you need to cite a retracted article, indicate the article’s retracted status in the References list and also include a citation and full reference for the retraction notice.

Additional Editor Comments:

Line #20: So, our aim was to assess.

Suggestion: make it; the aim of this study was to assess.

Line #23: An institution-based cross-sectional study involving 306 family caregivers of stroke patients selected through a consecutive sampling method were employed.

Suggestion: an institution-based cross-sectional study of 306 family caregivers of stroke patients was conducted using a consecutive sampling method.

Line #24: An interviewer- administered technique using a structured questionnaire was conducted.

Suggestion: A structured interviewer-administered questionnaire was used to collect data.

Line #39: This study found that the prevalence of depression among family caregivers of stroke patients was almost in similar range with the pooled prevalence of world report.

Suggestion: I suggest that it is important to present the meaning of your findings in the context of your study. This means explaining what your findings mean for the participants in your study, and for the broader field of research.

As your study focuses on a specific population, it is not appropriate to make general comparisons to the global context.

Line #80: So, our study was aimed to assess the prevalence and associated factors of depression among study participants at Jimma Medical Center.

Suggestion: This study aimed to assess the prevalence and associated factors of depression among family caregivers of patients with stroke at Jimma Medical Center.

Line #182 and 195: Nearly two-third, 64.7% (n=198) of family caregivers and more than half,

182 54.2% (n=166) of stroke patients were males. (See table 1 below). Among users of khat, alcohol, and tobacco over the past three months, about 29.3% (n=48), 29.4%(n=50) and 44.9%(n=61) were found to have depression respectively. (See table 2 below)

Consider my former comment and make it (table 1) and (table 2).

Line #251; UK university of 251 Glasgow (29)(30).

Suggestion: consider merging the citations. Apply the same format to line 256; this is supported by study done in Japanese (27)(32)

Line #275: As it was interviewer administered social desirability bias might be evident and being facility-based study may decrease the generalizability of the result of the finding.

Suggestion: consider re-writing as since the study was interviewer-administered, social desirability bias was a potential concern. Additionally, the facility-based nature of the study may limit its generalizability.

Line #276: Some of the information was gathered from the patient's hospital case folder, which is secondary data.

Yes, using secondary data can be a limitation, but it is not necessarily a limitation in and of itself. The quality and reliability of secondary data can vary broadly. Instead, it is good to consider some of the potential limitations of using secondary data in research including:

- The data may not have been collected in a way that is fitting for your research question.

- The data may be incomplete or inaccurate.

- The data may be biased.

- The data may not be generalizable to your population of interest.

---

## [Author Response · Author response to Decision Letter 5]

21 Sep 2023

Reviewer # 4 and academic editor

1. Please review your reference list to ensure that it is complete and correct. If you have cited papers that have been retracted, please include the rationale for doing so in the manuscript text or remove these references and replace them with relevant current references.

Thank you for your constructive and on time comments. T

he comment was accepted and corrected accordingly.

2. comments of academic editor regarding grammatical and structural correction of sentences.

That’s good and constructive comments.

We have corrected accordingly and it’s highlighted

3. Line #276: Some of the information was gathered from the patient's hospital case folder, which is secondary data.

Yes, using secondary data can be a limitation, but it is not necessarily a limitation in and of itself.

Actually, what we have gathered from patients’ folder was diagnosis, comorbid medical illness, type of stroke and other clinical factors. We put our concern as limitation only because of secondary data. otherwise, we didn’t face any missing data, gaps etc. so we have removed that we have considered as limitation previously.

---

## [Decision Letter · Decision Letter 5]

26 Feb 2024

Dear Dr. Jabessa,

Thank you for submitting your manuscript to PLOS ONE. After careful consideration, we feel that it has merit but does not fully meet PLOS ONE’s publication criteria as it currently stands. Therefore, we invite you to submit a revised version of the manuscript that addresses the points raised during the review process.

We look forward to receiving your revised manuscript.

Kind regards,

Nega Degefa Megersa, Msc

Academic Editor

PLOS ONE

Journal Requirements:

Additional Editor Comments:

Comments to authors

The document still contained some grammatical errors, typos, and punctuation mistakes. Here are some of the flaws:

Line # 24: capitalize “a” in the beginning of the sentence.

Line # 28: The association between depression and associated factors was estimated using an odds ratio at 95%CI and p-value of 0.05 on the final model. The sentence has grammatical problems. Modifications should be considered as follows:

The association between depression and associated factors was estimated using an odds ratio of 95%CI and a p-value of 0.05 in the final model.

Line # 31: conclusion and recommendation.

Look at the manuscript formatting and give a heading accordingly. For instance, there is no portion, namely the conclusion and recommendation. Instead, it is a conclusion.

Line # 84: from July 15–Sept 15, 2021. Make it September.

Line #88: Currently it provides service for more than 15 million populations.

Modify the sentence as follows: Currently, it provides service for over 15 million people.

Line #89: The study was conducted at a chronic disease follow up clinic, among family caregivers of stroke patients. The sentence has erroneous use of punctuation. Omit the comma and include a hyphen, as follows:

The study was conducted at a chronic disease follow-up clinic among family caregivers of stroke patients.

Line #115: The independent variables associated with caregivers include socio-demographic.

The independent variables in the study are sociodemographic variables,...

Though technically independent in nature, these factors were not specifically tested for their association with the outcome variable in this study. Instead of implying a potential relationship, it's best to simply refer to them as independent variables included in the research.

Reviewers' comments:

Reviewer's Responses to Questions

**Comments to the Author**

Reviewer #5: (No Response)

Reviewer #6: All comments have been addressed

2. Is the manuscript technically sound, and do the data support the conclusions?

Reviewer #5: (No Response)

Reviewer #6: Yes

3. Has the statistical analysis been performed appropriately and rigorously?

Reviewer #5: Yes

Reviewer #6: No

4. Have the authors made all data underlying the findings in their manuscript fully available?

Reviewer #5: Yes

Reviewer #6: Yes

5. Is the manuscript presented in an intelligible fashion and written in standard English?

Reviewer #5: No

Reviewer #6: Yes

Reviewer #5: General comment: The authors assessed the prevalence and associated factors of depression among caregivers of patients with stroke in South West Ethiopia. The authors used standardized measures to meet their objectives. They are commended for coming up with this study. The manuscript has a number of typos and grammatical errors. The Authors are advised to do a detailed proof reading before the manuscript is published.

Please see my specific comments below.

Abstract:

Is well written

Background.

Line 69 is not clear, please modify

Check line 72, 75 and 77 (grammatical errors and typos).

Line 78 that starts with “Worldwide” should have come earlier in the background.

Generally, the authors have tried to review the previous literature on the subject. However, the whole background lacks coherence and the rhetoric connection to create a compelling story for carrying out this study. The authors can do better.

Line 87 “Now a day it become under federal Ministry………This sentence makes no sense at all.

Line 94: what do the author mean by source of Population.

97 Whereas,” w should be capital

Line 1115: Caregivers were include?

The results and analysis appear to be well and good.

Reviewer #6: Reviewer comments to authors:

The authors structured the paper nicely to assess the prevalence and associated factors of depression among stroke family caregivers on follow-up at Jimma Medical Center, southwest Ethiopia. I found that this paper intends to fill a gap in the literature related to depression among stroke family caregivers on follow-up. However, I found some issues discussed below:

1. In the introduction:

- There are grammar and/or typing errors in your manuscript. For instance, on line 75, "hahigher..." Please correct it with “have a higher.”.

2. In the methods:

- There are many kinds of logistic regression models. I realized that you have used binary logistic regression. Thus, please replace logistic regression by binary logistic regression analysis in your manuscript.

3. In the results:

- On Table 2, have you presented the results of bivariable and multivariable binary logistic regression analyses together? What does it mean? Please correct the table. If it’s the result of multivariable binary logistic regression analysis, say, “Table 2: Multivariable binary logistic regression analysis for factors associated with depression among family caregivers of patients with stroke follow-up treatment at Jimma Medical Center, southwest Ethiopia, 2021.”.

- What does independent factor mean? Please say either independent variables or factors.

**Do you want your identity to be public for this peer review?** For information about this choice, including consent withdrawal, please see our Privacy Policy

Reviewer #5: No

Reviewer #6: No

---

## [Author Response · Author response to Decision Letter 6]

28 Mar 2024

We are grateful to the reviewers and academic editor for the time taken to review our manuscript, and the useful comments they provided to improve the manuscript. We have responded to the entire reviewer’ comments point-by-point, highlighting the changes in the manuscript; and they are summarized below: Once again, we thank the reviewers for the useful comments and suggestions.

Comments of the reviewers and editors

Comments raised by academic editor regarding typing, punctuation and grammatical error

Thank you for your constructive and on time comments. The comment was accepted and corrected accordingly.

Reviewer # 5 We have great thanks for the comments and the comment was accepted and corrected accordingly.

Comment regarding line # 78 was taken to the beginning of introduction part.

The background section has been modified as its highlighted.

Some typing and grammatical error has been corrected accordingly

Reviewer # 6 Thank you for the guiding comments you gave us. For the comments regarding to typing and grammatical error we tried to correct as highlighted.

We have used binary logistic regression analysis and it is mentioned throughout the document.

We presented the bivariable and multivariable result with single table, because it is recommended previously to merge the tables. To identify the result of each binary logistic analysis, its indicated separately by single and double “*” at the bottom of the table.

---

## [Decision Letter · Decision Letter 6]

21 May 2024

PONE-D-22-12677R6

prevalence and associated factors of depression among stroke family caregivers on follow up at Jimma medical center, southwest, Ethiopia: A cross-sectional study

PLOS ONE

Dear Dr. Jabessa,

Thank you for submitting your manuscript to PLOS ONE. After careful consideration, we feel that it has merit but does not fully meet PLOS ONE’s publication criteria as it currently stands. Therefore, we invite you to submit a revised version of the manuscript that addresses the points raised during the review process.

https://journals.plos.org/plosone/s/submission-guidelines#loc-laboratory-protocols . Additionally, PLOS ONE offers an option for publishing peer-reviewed Lab Protocol articles, which describe protocols hosted on protocols.io. Read more information on sharing protocols at https://plos.org/protocols?utm_medium=editorial-email&utm_source=authorletters&utm_campaign=protocols .

We look forward to receiving your revised manuscript.

Kind regards,

Nega Degefa Megersa, Msc

Academic Editor

PLOS ONE

Journal Requirements:

Additional Editor Comments:

Line 25: A structured interviewer-administered was used to collect data.

Do you mean a structured interviewer-administered questionnaire was used to collect data?

Line 28: The association between depression and associated factors was estimated using an odds ratio of 95%CI and a p-value of 0.05 in the final model.

Logistic regression is a common method to estimate the association between outcomes and independent variables. But it's important to note that a p-value less than 0.05 (not exactly 0.05) is typically used as evidence of a statistically significant association. Additionally, the strength of this association can be measured by an odds ratio, along with its 95% confidence interval.

Line 85 to 90: Although depression is very common among stroke caregivers, little is found on literature regarding to our country.

Generally, the caregivers’ are more vulnerable to develop emotional, physical related problems 88 and limited social life than the general population (14). This study was aimed to assess the 89 prevalence and associated factors of depression among family caregivers’ of patients with stroke 90 at Jimma Medical Center (JMC).

To strengthen your writing, consider combining the existing paragraphs into a single, unified one. This new paragraph should clearly explain the gap in current research (what's missing) and how your study aims to fill that gap (the purpose). Additionally, you might want to review the paragraphs for any grammatical errors to ensure clarity and readability.

Although depression is very common among stroke caregivers, little is found in literature regarding our country. Generally, caregivers are more vulnerable to developing emotional, physical related problems and limited social life than the general population (14). This study was aimed at assessing the prevalence and associated factors of depression among family caregivers of patients with stroke at Jimma Medical Center (JMC).

Line 112: sample size determination

To ensure adequate power for both objectives in your study,

Did you calculate the sample size needed to identify significantly associated factors for your second objective? Compare this sample size to the one calculated for your first objective? If the sample size required for the first objective is larger, then you should use the larger sample size for your entire study.

Consider clearly outlining these steps in your sample size determination.

Reviewers' comments:

Reviewer's Responses to Questions

**Comments to the Author**

Reviewer #7: (No Response)

2. Is the manuscript technically sound, and do the data support the conclusions?

Reviewer #7: Yes

3. Has the statistical analysis been performed appropriately and rigorously?

Reviewer #7: Yes

4. Have the authors made all data underlying the findings in their manuscript fully available?

Reviewer #7: Yes

5. Is the manuscript presented in an intelligible fashion and written in standard English?

Reviewer #7: No

**Reviewer #7:**  Dear editor,

Thank you very much for giving me the chance to read and review this important manuscript. Overall, the authors reported important findings, but the English language and grammar should be improved.

Abstract:

“A structured interviewer-administered (What?)…………was used to collect data” Please improve this sentences.

“The Patient Health Questionnaires-9 (PHQ-9) was used to screen outcome variables (i.e., depression)” It's good to say variable instead of variables since your outcome variable one/depression.

“This study found that the prevalence of depression among family caregivers of stroke patients was 35.9% and almost the same with similar study design conducted across different countries of the world.” Please remove 35.9% here and modify the sentences. In addition, could you please write a specific and applicable recommendation (who will be responsible for detecting it early)?

Introduction

Overall, it needs proofreading in language and grammar. “Depression has the greatest decrement in the health status of the population either alone or in conjunction with other chronic medical illnesses such as Asthma, Angina, and/or diabetes.” Please make this sentence clear and meaningful.

“Most of the time, caregivers have been considered as one of the overlooked patients and this could result in poor health outcomes among caregivers (10).” Caregivers are not patients; it is also an offensive word. Please modify it.

Please explain the gap in the study since there are multi center studies in Ethiopia, for example https://doi.org/10.2147%2FNDT.S418074). Your study is limited to an institution and a specific place. What is new about this study?

Methods

What is the reason behind including caregivers who attend for two weeks or more?. Do you believe the degree of depression is similar between the different groups of family members, for example, the mother and other relatives?

Why did you use the Nigerian study to calculate the sample since there is a study in Ethiopia?.

Result and discussion:

“From the total family caregivers about 46.4%(n=142), 44.4%(n=136) and 44.4% (n=136) of them were used khat, tobacco and alcohol over the past three months respectively” Please see your data again for this report, since you are reporting that 44.4% of caregivers are smokers.

Please proofread the grammar in your discussion. While interpreting your findings with other literature, please give clear justification for why your study is lower or higher than other studies. For example, consider how socioeconomic differences between countries contributed to depression. How using PHQ-9 and BDI varies on prevalence of depression, etc.

“The other supporting reason might be in that, the study was conducted among family caregivers of stroke patient whose illness duration was less than six months in which the depression status increased during the acute phase of the illness(26).” Modify sentences and no need of coma.

Conclusion

“This study found that the prevalence of depression among family caregivers of stroke patients 295 was almost in similar range with the pooled prevalence of world report” Why do you compare it with a global study since there are local studies? If there is a pooled study, what is the value of doing this single study?

**Do you want your identity to be public for this peer review?** For information about this choice, including consent withdrawal, please see our Privacy Policy

Reviewer #7: **Yes: ** Alemayehu Molla Wollie

---

## [Author Response · Author response to Decision Letter 7]

27 Jun 2024

Comments of the reviewers and editors

Comments raised by academic editor on spelling and grammatical

Thank you for your constructive and on time comments.

The comment was accepted and corrected accordingly.

Regarding sample size calculation we didn’t calculate sample size for associated factors considering that it’s optional.

This can be taken as a limitation

Reviewer # 7

Dear reviewer we have great thanks for the comments you have raised and we try to address one by one.

Regarding to spelling error the comment was accepted and corrected accordingly.

Regarding the prevalence the previous reviewers commented me to include the prevalence in abstract part.

Comments in introduction part regarding abusive words and grammatical error was corrected accordingly

Methodology

#Regarding this study, up to this research was conducted no published research conducted in our country regarding this title, thus being the first finding for this country may fill the gaps in our country and that is why we used the Nigerian finding.

#Even most of the researches done outside the country were conducted before five years.

#The reason why we include the caregivers who at least spent two weeks was; in order to diagnose depression at least two weeks is mandatory, even if this is for screening purpose.

# Kinship relation was one of the factors that we have used as independent factor and associated with depression at bivariable binary logistic regression, but fail to associate at final model.

Reviewer # 7 Discussion

We accepted the comments and try to address and modify the discussion parts

---

## [Editor Report · Decision Letter 7]

1 July 2024

prevalence and associated factors of depression among stroke family caregivers on follow up at Jimma medical center, southwest, Ethiopia: A cross-sectional study

PONE-D-22-12677R7

Dear Dr. Jabessa,

We’re pleased to inform you that your manuscript has been judged scientifically suitable for publication and will be formally accepted for publication once it meets all outstanding technical requirements.

Kind regards,

Nega Degefa Megersa,  

Academic Editor

PLOS ONE
---

## [Editor Report · Acceptance letter]

PONE-D-22-12677R7

PLOS ONE

Dear Dr. Jabessa,

I'm pleased to inform you that your manuscript has been deemed suitable for publication in PLOS ONE. Congratulations! Your manuscript is now being handed over to our production team.

Kind regards,

on behalf of

Mr. Nega Degefa Megersa

Academic Editor

PLOS ONE